# Muscle Tone and Body Weight Predict Uphill Race Time in Amateur Trail Runners

**DOI:** 10.3390/ijerph18042040

**Published:** 2021-02-19

**Authors:** César Berzosa, Héctor Gutierrez, Pablo Jesús Bascuas, Irela Arbones, Ana Vanessa Bataller-Cervero

**Affiliations:** Facultad de Ciencias de la Salud, Universidad San Jorge, Autov. A-23 Zaragoza-Huesca, km 299, 50830 Villanueva de Gállego, Zaragoza, Spain; cberzosa@usj.es (C.B.); pbascuas@usj.es (P.J.B.); iarbones@usj.es (I.A.); avbataller@usj.es (A.V.B.-C.)

**Keywords:** fatigue, vertical impacts, stiffness, GPS

## Abstract

Background: Vertical kilometer is an emerging sport where athletes continuously run uphill. The aims of this study were to assess changes in vertical impacts caused by uphill running (UR) and the relation between the anthropometric and lower limb muscular characteristics with speed. Methods: Ten male experienced runners (35 ± 7 years old) participated in this study. In the racetrack (4.2 km long, 565 m high), seven sections were stablished. Mean speed and impact value of sections with similar slope (≈21%) were calculated. The gastrocnemius stiffness (GS) and tone (GT); and the vastus lateralis stiffness (VS) and tone (VT) were assessed before the race. Results: Pearson’s correlation showed a linear relationship between vs. and VT (*r* = 0.829; *p* = 0.000), GT and GS (*r* = 0.792; *p* = 0.001). Mean speed is correlated with weight (*r* = −0.619; *p* = 0.024) and GT (*r* = 0.739; *p* = 0.004). Multiple linear regressions showed a model with weight and GT as dependent variables of mean speed. Mean impacts decreased significantly between sections along the race. Conclusions: The vertical impacts during UR were attenuated during the race. Moreover, body weight and GT were associated with the time-to-finish, which supports that low weight alone could not be enough to be faster, and strength training of plantar flexors may be a determinant in UR.

## 1. Introduction

Trail running or mountain running is a sport that has become very popular in the last years, and the research associated with this long run discipline has also increased [1,2,3,4]. One of the modalities of trail running is the vertical kilometer, where the whole track is uphill. In this type of event, athletes must complete an uphill route of 1 km vertical elevation increase. In addition, a minimum average of 20% positive slope must exist and one or more sections of 5% positive slope of the total distance race must be included. Regarding the length and type of terrain, the maximum length must be 5 km and the terrain can vary between different races [5].

Uphill running is a very demanding activity. The athlete must perform positive mechanical power in order to displace their body upward against gravity. This generated mechanical power increases with increasing slope.

Compared to level running, in uphill running, due to the lower limb position, hip joint muscles increase their work, whereas the work of knee and ankle remains similar than in level running [6]. The use of elastic energy also changes during uphill running. While most of the energy stored in tendons is recovered in level running [7], running uphill with steeper slopes entails the necessity of raising the center of mass, causing an increase in the positive network generated by the body, considering that the elastic energy stored cannot be used due to the increase in ground contact time. Analyzing the mechanical efficiency, previous studies have proven than uphill runners show around 25% of efficiency, this value corresponds only to muscle contraction [8,9]. Uphill running cost could explain running performance because this parameter differs from level running. Balducci et al. (2016) [10] studied the influence of stride length, stride frequency, and body mass index, and no correlation was found between these parameters with the cost of uphill running compared to level running. Nevertheless, influence of these anthropometric measures was not studied directly in uphill runners. For example, body weight has been shown to be a predictor in the performance of marathon runners [11] and it seems plausible that this factor also influences the performance of a modality in which it is necessary to raise your body on every step.

During running, the foot strikes the ground decelerating the body to zero and generating large ground reaction forces (GRF) [12]. Step by step, impacts are transmitted through the musculoskeletal system. Both passive and active mechanisms act in order to attenuate the shock, minimizing the damage. Although running impacts do not reach extreme values, the quantity of running impacts can be significant. Impact peak is very sensitive to leg stiffness and the damping effect of foot (pads) and leg [13].On one hand, the impacts received by the body in each step are attenuated by the bone bending, heel pads, and intervertebral discs as passive dampers; on the other hand, lower limb muscles work actively in order to absorb the impacts.

In order to assess running impact acceleration magnitude, several methods could be used, one of them is the tibial acceleration peak, assessed through the placement of an accelerometer in the shank [14], whose value is approximately 8 g [15]. Another method could be accelerometry measurement in the sacrum, presenting smaller values [16]. Through these measurements, the role of active and passive mechanisms to reduce the impact received during running from the shank to the head could be determined [17].

Several studies have analyzed the effect of fatigue on impacts, but their results have not offered a clear conclusion [18]. The possible cause of these discrepancies may be the existence of differences in the way of assessing running impacts between studies. Research articles that have analyzed the ground reaction forces disagree on the effect that fatigue has on running impacts [19], finding both increases [20] and decreases [21], and offering different explanations for it. The rise could be explained by an increased lower limb stiffness [22]. In addition, another study found a correlation between the pre-activation of gastrocnemius and GRF [23]. The regulation of the lower limb stiffness and the reduced storage of elastic energy could be a possible explanation for the decrease in GRF due to fatigue [24].

Researchers that measured the impacts with accelerometers found an increase in peak impact due to fatigue [25]. The different strategies used to face the fatigue could influence the responses. In long-distance runners, the global fatigue increases the impacts as well as the local fatigue due to the muscle activity imbalance [26]. Changes in lower limb stiffness can also have an influence in peak accelerations [14] as well as running technique. Crowell and Davis (2011) [27] found a relationship between running technique and the impacts.

The objective of this study was to assess the relationship between anthropometric and the lower limb muscular characteristics with speed, and to analyze the changes in vertical impacts caused by uphill running.

## 2. Materials and Methods

### 2.1. Participants

Ten male recreationally trained runners participated in the study (age 35 ± 7 years old, mass 68.2 ± 5.3 kg, height 1.77 ± 0.03 m, BMI 21.6 ± 1.4 kg/m^2^). The inclusion criterion were, at least, one year of experience in trail running races and not suffered from lower limb injuries in the last three months. All the subjects were participants in the uphill race “TurrónSkyrace Pico Las Calmas”, celebrated in Arguis, Huesca (Spain). The participants signed a written consent previous tothe data collection and this study was approved by the Ethics Committee of the University. All procedures followed the Declaration of Helsinki on the use of human subjects.

### 2.2. Procedure

The uphill race was 4.2 km long and had a 565 m positive slope. In this time-trial race, the runners started to run every 30 s and attempted to complete the racetrack as quickly as possible. In the racetrack, seven sections were distinguished, defined by six control points (Figure 1). For each section, the mean speed of the runners was calculated from the time in each control point and the official distance track. The GPS device was synchronized to the official race start time by means of the detection of the first change in velocity after a long period of standing.

The measurements of the skeletal muscle tone and stiffness were taken in a tent provided by the organizer, near the race start zone.

### 2.3. Measurements

Muscle stiffness and tone were collected in lying position (both prone and supine position depending on the muscle evaluated) by a hand-held myometer (Myoton-Pro, Myoton AS, Tallinn, Estonia). Surface Electromyography for the Non-Invasive Assessment Muscle (SENIAM) guidelines [28] were followed to draw on the skin at the testing locations. Medial head of gastrocnemius (MG), lateral head of gastrocnemius (LG), vastus lateralis (VL) tone and stiffness were analyzed in the relaxed position. The probe of the Myoton-Pro was placed perpendicular to the skeletal muscle surface in each measurement. Five consecutive measurements were taken at each site, giving the mean stiffness in N/m and the mean tone measured in Hz [29].To ensure validity of the data, a measurement with a coefficient of variation fewer than 3% was accepted, and any measurement above this value was rejected and measured again. The Myoton-Pro offers good to excellent test-retest reliability for lower body tone and stiffness assessment [30].

Muscle tone is calculated using the following formula:(1)F=fmax

Muscle stiffness is calculated using the following formula:(2)$$S=\frac{amax\times mprobe}{\Delta l}$$

Raw data were grouped together in order to mitigate the effect of the asymmetries between both limbs (left and right) and to consider possible synergies between muscles (the medial and lateral gastrocnemius, the vastus medialis, and lateralis). Four parameters of the participants’ muscle mechanical properties emerged out of this procedure: the gastrocnemius muscle stiffness (GS), the gastrocnemius muscle tone (GT), the vastus lateralis stiffness (VS), and the vastus lateralis tone (VT). Body weight was assessed using a Tanita BC-1000 scale.

The runners wore a vest during the race with a pocket located in the back (at the height of vertebrae C7) where an Apex GPS device (STATSport Group, Newry, Ireland, UK) was placed. This device includes a 18 Hz GPS and 100 Hz accelerometer in three axes, 100 Hz gyroscope, and 10 Hz magnetometer.

The accelerometer signal of the first 30 s of every race section was evaluated in a custom MAATLAB routine (The Mathworks Inc, Natick, MA, USA) in order to analyze the vertical impact acceleration (VIA) in each slope section. The magnitude of the accelerometer signal was low-pass filtered with a fourth order Butterworth filter with a cut-off frequency of 10 [16]. The mean value of the 15 first peaks, corresponding to the first 15 steps was calculated [31]. The mean impacts of sections with similar slope were compared in order to check the fatigue effect changes on the impact magnitude as a marker of fatigue. In this way, Sections 2, 4, 5, and 7 (Table 1) were analyzed (VIA_1, VIA_2, VIA_3, VIA_4, respectively).

Aside from the impacts, the mean speed (MS) of each section was obtained from the GPS devices’ software (Apex Software, STATSport Group, Newry, Ireland, UK).

### 2.4. Statistical Analysis

Statistical analyses were performed using SPSS version 21.0 for Windows (SPSS Inc., Chicago, IL, USA). Descriptive statistics mean, standard deviation (SD), lower 95% confidence limit (LCIL95%) and upper 95% confidence limit (UCIL95%) were calculated for weight, mean speed, VIA, GS, VS, GT, and VT. Normality of datasets was checked with the Shapiro–Wilk test. Pearson’s correlation was calculated and used to determine lineal relationships between all measures.

A repeated measures one-way ANOVA were performed to compare VIA along the race (VIA_1, VIA_2, VIA_3, VIA_4) for all participants. The W de Mauchly test was used as sphericity criteria. Regression model (lineal, quadratic or cubic) with the highest order among all those that presented statistical significance was considered as the ideal model [32]. Eta square value (η^2^) was used for effect size calculation. Bonferroni’s post-hoc procedure was applied to locate pair-wise differences [33].

Multiple linear regressions were calculated using a “stepwise” method. Mean speed was considered the dependent variable and weight, VIA, GS, VS, GT, and VT as possible independent variables. Entry and exit criteria were an F probability greater than 0.05 and 0.10, respectively. Residual linearity and independence assumptions were checked with the Durbin–Watson test; values between 1 and 3 in the Durbin–Watson test were considered an acceptable criterion. Homoscedasticity was studied in a standardized residual-standardized prediction plot. Normality of residuals was checked with the Shapiro–Wilk test. Multicollinearity was estimated by a variance inflation factor (VIF), values greater than 10 were considered as excessive multicollinearity. Cases with Cook’s distance greater than 1 were indicated as influential cases and removed fromthe data analysis [34].

All tests were performed with a level of significance of *p* < 0.05.

## 3. Results

A summary of the descriptive statistics of the sample can be studied in Table 2.

Pearson’s correlation (Table 3) showed a linear relationship between VT and vs. (*r* = 0.829; *p* = 0.000), and GT and GS (*r* = 0.792; *p* = 0.001). Furthermore, mean speed was correlated with weight (*r* = −0.619; *p* =0.024) and GT (*r* = 0.739; *p* = 0.004).

Variables included in the repeated measures one-way ANOVA showed an adequate sphericity (W de Mauchly = 0.546; *p* = 0.263). Only a linear model presented statistical significance (*p* = 0.000). Quadratic (*p* = 0.193) and cubic (*p* = 0.758) could not be considered. The effect size of the time factor over vertical impact acceleration was η^2^ = 0.621. Bonferroni’s post-hoc procedure revealed mean differences (MD) statistically significant between VIA_1 toVIA_2 (MD = 0.429; *p* = 0.006), VIA_1 to VIA_3 (MD = 0.688; *p* = 0.005), VIA_1 to VIA_4 (MD = 0.891; *p* = 0.000), andVIA_2 toVIA_4 (MD = 0.462; *p* = 0.015). These differences can be observed in Figure 2.

Multiple linear regressions showed a model with weight and GT as dependent variables of mean speed (Equation (3)). This model presented a R^2^ = 0.732 and an adjusted R^2^ = 0.678 (*p* = 0.025). Model predictive capacity was statistically significant (*p* =0.001). The model’s coefficients were statistically significant (A = 0.179, *p* = 0.05; B = −0.028, *p* = 0.025); however, the model’s constant was not (C = 0.907, *p* = 0.468).Standardized coefficients were A_beta_ = 0.615 and B_beta_ = -0.448,respectively.
(3)MS=0.179×GT−0.028×weight+0.907

A Durbin–Watson test value of 2496 confirmed residual linearity and independence assumption. The standardized residual–standardized prediction plot did not show any relationship confirming homoscedasticity. The Shapiro–Wilk test confirmed normality of the residuals (*p* = 0.221). Multicollinearity between the dependent variables was not observed (VIF = 1.084). Excessive influential cases were not observed (max Cook’s distance = 0.819; min Cook’s distance = 0.004).

## 4. Discussion

The main objective of this study was to analyze the relationship between anthropometric and muscular characteristics of the lower limb with the speed in an uphill running race, while the second aim was to assess changes in the vertical impacts caused by uphill running. The main findings of this study were as follows: (1) There was a positive correlation in mean speed and gastrocnemius muscle tone and between tone and stiffness of leg and thigh muscles, and (2) a progressive decrease in vertical impacts along the race was observed.

In addition, an inverse correlation between race speed and weight was found. It has been observed that faster athletes weighed less. According to that, a study with marathon runners showed that race speed was correlated with body fat, but not with the total weight [11]. Contrary to what happens in level races, uphill runners have to raise their own weight step by step, which could explain these controversial results.

According to the multivariate model including weight and gastrocnemius tone, 68% of the variance in race speed could be explained. This finding suggests the importance of these variables in uphill running performance, although other parameters as metabolic values (e.g., oxygen uptake) [35] or tone of different muscles (e.g., gluteus maximus) [36] could be very important and explain part of the other 27% of the variance in race times. The predicted race speeds were explained by only two variables (weight and GT), showing a strong predictive capacity (*p* = 0.001) and R^2^ = 0.73 with the real registered race times. This equation should be validated in a different group of participants and could be applied only to runners with the same characteristics (age, gender, years of experience, and training status).

Weight is the first of the parameters to consider in these types of races. It could seem obvious that, if you must lift your body to the top of a mountain, the smaller your weight, the faster you are. Saunders et al. (2004) [37] found an inverse correlation between running economy and body weight, with the lightest runners also being the most economical. Pate et al. (1992) [38] studied several variables (physiological, anthropometric, and training load) and their relationship to running economy, showing that the lightest runners were the fastest. Running economy is a key factor in running performance, obtaining better race times, as has been reported in this study. However, another study observed that there was no relationship between these aspects [11,39,40]. Perhaps this disagreement could be due to the differences in the level of body fat of runners. Lower levels of body fat are a variable predictive of running performance in terms of running speed [11] or performance in treadmill tests [41].

The gastrocnemius muscle tone is the second of the variables included in the predictive model. It is the only parameter of the muscle properties that fit in the multivariate regression analysis, although other parameters are clearly related to it. In this way, the GT and GS are significantly correlated as well as the VT and VS. All these facts could be explained, at least partially, because there is a higher ankle joint moment and a decrease in the knee moment associated with increasing slope [6]. The higher moment in ankle joint is mainly produced by gastrocnemius muscle action, and it is in accordance with the correlation found between gastrocnemius muscle tone and mean velocity during the running race segments analyzed. In another study, it was also described that stiffness is important in activities like jumping or sprinting [29], and according to the results found, it could also be important during uphill running. As previously mentioned, there is a positive correlation between muscle tone and stiffness, both in gastrocnemius and vastus medialis. Maybe training muscle tone and stiffness through strength training could be beneficial in order to improve uphill running performance.

Related to strength training, it could also be important in order to increase the attenuation of VIA along the race when comparing sectors with similar slopes. These results are quite interesting and suggest that during uphill running, the neuromuscular behavior is similar to previous studies [18]. Other studies have shown increases or similar vertical impacts and Zadpoor and A Nikooyan (2012) [19] concluded in their review that no significant changes could be found in ground reaction forces due to fatigue during running, although they reported studies that showed increases and decreases in GRF. It is also interesting that while level running increases vertical impacts possibly because of the change to rearfoot running strike pattern [25], in our study, VIA is attenuated while the running race is going on. These results are similar in both fast and slow athletes, so VIA attenuation is not related to race speed as could be thought, and the reduction in vertical impacts is common, so more studies should be performed to clarify this aspect. In our study, a slower speed in T2 (related to VIA1) was also observed than in T4 and T5 (related to VIA2 and 3). This could be explained due to the characteristics of the terrain as T2 is a canyon, while T3 and T4 are paths and are easier terrain to run.

## 5. Conclusions

In conclusion, the vertical impacts generated during uphill running are attenuated during the race, maybe because of the fatigue accumulated and the running technique changes linked to it. Even more, body weight and gastrocnemius tone were associated with the time-to-finish during an uphill running race in amateur athletes. It supports the idea that a greater muscle tone in ankle plantar flexors and a low body weight are determinant to achieve performance in these kinds of trail running races. Low body weight alone could not be enough to reach a faster race time, maybe accompanying body weight control with a strength training program focused on plantar flexors, emphasizing the eccentric and plyometric work to increase muscle stiffness may be determinant in uphill running.

## Figures and Tables

**Figure 1 ijerph-18-02040-f001:**
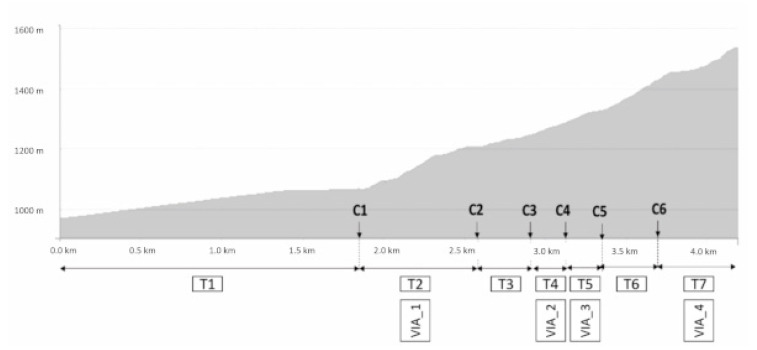
Track of the race with control points (C1–C6), sections (T1–T7), and vertical impact acceleration (VIA_1-VIA_4).

**Figure 2 ijerph-18-02040-f002:**
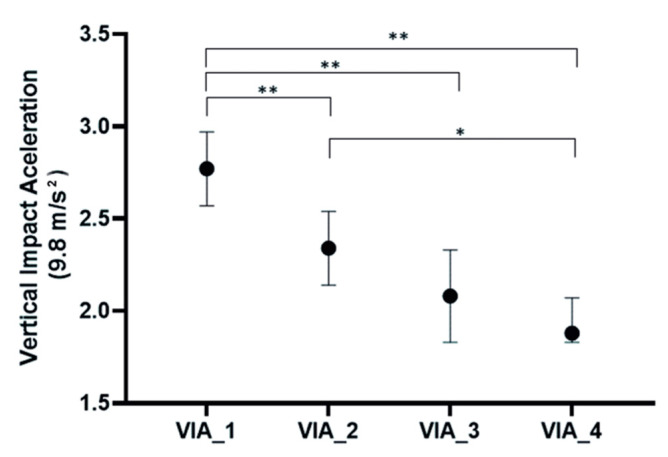
Vertical impact acceleration in four consecutive sectors with a similar slope of a vertical race. VIA: Vertical impact acceleration. *****
*p* < 0.05; ******
*p* < 0.01 in Bonferroni’s post-hoc test.

**Table 1 ijerph-18-02040-t001:** Slope and horizontal distance of every section of the race besides the type of surface.

	T1	T2	T3	T4	T5	T6	T7
Distance (m)	1800	800	250	200	220	390	380
Slope (%)	11	22	11	20	21	34	23
Terrain	Track	Canyon	Path	Path	Path	Path	Track/path

**Table 2 ijerph-18-02040-t002:** Descriptive statistics of the participants. Standard deviation (SD), lower 95% confidence limit (LCIL95%); upper 95% confidence limit (UCIL95%), vertical impact acceleration (VIA),mean speed (MS), finishing time (min), gastrocnemius stiffness (GS), gastrocnemius tone (GT), vastus lateralis stiffness (VS), vastus lateralis tone (VT).

N = 13	Mean	SD	LCIL 95%	UCIL 95%	*p*
MS (m/s)	1.81	0.30	1.63	1.98	0.168
Weight (kg)	68.85	4.79	65.95	71.74	0.140
Finishing Time(min)	37.2	6.20	33.45	40.95	0.168
VIA_1 (9.8 m/s^2^)	2.77	0.34	2.57	2.97	0.570
VIA_2 (9.8 m/s^2^)	2.34	0.34	2.14	2.54	0.299
VIA_3 (9.8 m/s^2^)	2.08	0.41	1.83	2.33	0.099
VIA_4 (9.8 m/s^2^)	1.88	0.32	1.69	2.07	0.099
VT (Hz)	14.80	1.71	13.77	15.84	0.787
GT (Hz)	15.65	1.01	15.04	16.26	0.945
VS (N/m^2^)	284.87	28.42	267.69	302.04	0.267
GS (N/m^2^)	280.38	15.09	271.26	289.50	0.211

**Table 3 ijerph-18-02040-t003:** Pearson’s correlation calculated to determine lineal relationships between all measures. Vertical impact acceleration (VIA), mean speed (MS) (m/s), gastrocnemius stiffness (GS) (N/m^2^), gastrocnemius tone (GT) (Hz), vastus lateralis stiffness (VS) (N/m^2^), vastus lateralis tone (VT) (Hz)*****
*p* < 0.05; ******
*p* < 0.01.

	Weight	VIA_1	VIA_2	VIA_3	VIA_4	GT	VT	GS	VS
MS	−0.619 *	0.169	0.198	0.200	0.041	0.739 **	0.006	0.483	0.231
Weight		−0.288	−0.485	−0.370	−0.484	−0.279	0.063	−0.030	−0.095
VIA_1			0.423	−0.114	0.256	0.031	0.035	−0.201	0.162
VIA_2				0.113	0.113	0.106	0.460	−0.354	0.365
VIA_3					0.544	0.233	0.082	0.248	−0.010
VIA_4						−0.117	−0.432	0.021	−0.450
GT							0.265	0.792 **	0.356
VT								−0.008	0.829 **
GS									0.104

## Data Availability

The data presented in this study are available on request from the corresponding author. The data are not publicly available due to legal and privacy issues.

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
