# Peer review of "Muscle Tone and Body Weight Predict Uphill Race Time in Amateur Trail Runners"

_ijerph, 2021, doi:10.3390/ijerph18042040_

Round 1

Reviewer 1 Report

I enjoyed your article and have an interested in ultra-endurance. The vertical kilometre is an interesting race / concept.

Overall the manuscript is written well in terms of content, but there are numerous spelling and grammatical errors. I started pointing these out, but there are quite a lot so I would ask that the manuscript gets checked thoroughly

 to correct these. Unfortunately, currently, they distract from the content.   

Abstract:

Line 20: …, which supports that

Line 21: may be a determinant

Introduction:

A very technical introduction. Although weight is listed as one of the measures, it is not addressed here. Could you please add this as it plays a vital role in uphill running performance.

In addition, the event itself is not described. Please add some more information about the type of events, the demands and duration (distances, percent slopes etc.).

Line 31: what does ‘the athlete must perform positive work’ mean?

Line 32: re-phrase. The amount of work performed increases, it isn’t higher.

Line 34: compared to, not ‘comparing to’

Line 35: increase their work not ‘its’

Line 53: dampers? Not ‘dumpers’

Methods:

Please state that this was a race and that the runners attempted to cover the distance as quickly as possible.

How was their weight assessed?

Results:

Please include the finishing time. Did you record any other variables, such as heart rate or similar? Please display these if yes.

Discussion:

You discuss bodyfat; did you measure this?

It would be beneficial to include some more information about the race. You mention speed among other things, and it would be useful to show this if possible.

The practical implications and applications should be expanded.  

Reviewer 2 Report

  • I think that the study is suitable for a sports physiology journal or similar, but I do not understand the link with Environmental Research and Public Health, nor the innovation for the scientific field of study, so I do propose that the author should frame the issue with the magazine or look for another journal.
  • Line 76 (ref 24)
  • In the objectives, you talk about fatigue, but do not evaluate fatigue (eg gas analyzer or lactate). There must be a relationship between the objectives and the conclusion, in the conclusion, there is talk of fatigue and technique, but in the objectives, only fatigue is mentioned and in the whole study none of these parameters were evaluated.
  • line 104 lying but... SUPINE or PRONE position?
  • in the measurements, it would be interesting to put the formulas of how they got to Muscle Stiffness and Muscle Tone
  • line 109 Why the average of 3 repetitions and not 5?
  • Table 2 change Shapiro Wilk to p only
  • Lines 181 to 184. The data described does not correspond to the table.
    VT to VS and GS to GS I think they are exchanged. MS to GT, not VT. Confirm in abstract.
  • Lines 197 to 199. The text is difficult to understand. Put it more Easier. "VIAx to VIAy (xxx), VIAx to VIAy (xxx) and VIAx to VIAy (xxx)"
  • Figure 2. If possible, place T2, T4, T5, and T7 under VIAs.
  • Line 211 isolate the equation to make it stand out (it's the strength of the investigation)
  • The discussion should take into account the characteristics of the terrain in the intervals and relate them to the data obtained
  • Line 275 put a reference 
  • Lines 278 to 280 fatigue and techniques not evaluated and you say in the discussion that the results are contradictory when compared to other studies (Fatigue).
  • Methodologically you selected the intervals with similar slopes, but the distances are variable by interval, mainly in T2 with more 600m than T4 and T5 and 480m than T7. I think that there should be a paragraph that justifies this decision, relating it not only to the slope but also to the distances in each interval.

Reviewer 3 Report

The authors took up a very interesting issue related to mountain running, with particular emphasis on muscle tone and body weight in order to predict the time of an uphill run. The job is very good. It was written in accordance with all the requirements of a scientific dissertation. It contains all the elements necessary for a reliable scientific research.Appropriate statistical methods were used and the results were presented in a logical and transparent manner. An excellent introduction introduces a detailed description of the issue and, combined with a discussion of the appropriate length, creates an interesting work not only for a small group of specialists, but also for the average reader, especially a runner, interested in developing his scientific knowledge.

Round 2

Reviewer 2 Report

The results presented in the text are different from the results presented in the summary
you must correct the summary

Author Response

The response is in the cover letter. Sorry for the mistake.
